# Early Respiratory Impairment and Pneumonia after Hybrid Laparoscopically Assisted Esophagectomy—A Comparison with the Open Approach

**DOI:** 10.3390/jcm9061896

**Published:** 2020-06-17

**Authors:** Martin Reichert, Maike Lang, Matthias Hecker, Emmanuel Schneck, Michael Sander, Florian Uhle, Markus A. Weigand, Ingolf Askevold, Winfried Padberg, Veronika Grau, Andreas Hecker

**Affiliations:** 1Department of General, Visceral, Thoracic, Transplant and Pediatric Surgery, University Hospital of Giessen, Rudolf-Buchheim Strasse 7, D-35392 Giessen, Germany; maike@lang.cd (M.L.); ingolf.askevold@chiru.med.uni-giessen.de (I.A.); winfried.padberg@chiru.med.uni-giessen.de (W.P.); andreas.hecker@chiru.med.uni-giessen.de (A.H.); 2Department of Pulmonary and Critical Care Medicine, University of Giessen and Marburg Lung Center (UGMLC), University Hospital of Giessen, Klinikstrasse 33, D-35392 Giessen, Germany; matthias.hecker@innere.med.uni-giessen.de; 3Department of Anesthesiology, Intensive Care Medicine and Pain Therapy, University Hospital of Giessen, Rudolf-Buchheim Strasse 7, D-35392 Giessen, Germany; emmanuel.schneck@chiru.med.uni-giessen.de (E.S.); michael.sander@chiru.med.uni-giessen.de (M.S.); 4Department of Anesthesiology, Heidelberg University Hospital, Im Neuenheimer Feld 110, D-69120 Heidelberg, Germany; florian.uhle@uni-heidelberg.de (F.U.); markus.weigand@uni-heidelberg.de (M.A.W.); 5Laboratory of Experimental Surgery, German Centre for Lung Research (DZL), Justus-Liebig-University Giessen, Feulgen Strasse 10-12, D-35392 Giessen, Germany; veronika.grau@chiru.med.uni-giessen.de

**Keywords:** abdomino-thoracic esophagectomy, Ivor Lewis esophagectomy, respiratory complication, Horovitz index, oxygenation, pulmonary function, pulmonary function index, pneumonia, lung injury, vagal nerve

## Abstract

Patients undergoing esophageal cancer surgery are at high risk of developing severe pulmonary complications. Beneficial effects of minimally invasive esophagectomy had been discussed recently, but the incidence of perioperative respiratory impairment remains unclear. This is a retrospective single-center cohort study of patients, who underwent open (OE) or laparoscopically assisted, hybrid minimally invasive abdomino-thoracic esophagectomy (LAE) for cancer regarding respiratory impairment (PaO_2_/FiO_2_ ratio (P/FR) < 300 mmHg) and pneumonia. No differences were observed in the cumulative incidence of reduced P/FR between OE and LAE patients. Of note, until postoperative day (POD) 2, P/FR did not differ among both groups. Thereafter, the rate of patients with respiratory impairment was higher after OE on POD 3, 5, and 10 (*p* ≤ 0.05) and tended being higher on POD 7 and 9 (*p* ≤ 0.1). Although the duration of LAE procedure was slightly longer (total: *p* = 0.07, thoracic part: *p* = 0.004), the duration of surgery (Spearman’s rank correlation coefficient (r_sp_) = −0.267, *p* = 0.006), especially of laparotomy (r_sp_ = −0.242, *p* = 0.01) correlated inversely with respiratory impairment on POD 3 after OE. Pneumonia occurred on POD 5 (1–25) and 8.5 (3–14) after OE and LAE, respectively, with the highest incidence after OE (*p* = 0.01). In conclusion, respiratory impairment and pulmonary complications occur frequently after esophagectomy. Although early respiratory impairment is independent of the surgical approach, postoperative pneumonia rate is reduced after LAE.

## 1. Introduction

Esophageal carcinoma is a common cancer with a growing incidence over the past years and a major cause for cancer-related mortality worldwide [1]. Subtotal resection of the esophagus also as part of a multimodal treatment strategy is the gold-standard for early stage cancer and for selected locally advanced tumors [1]. If the tumor is localized in the thoracic part of the esophagus, the abdomino-thoracic Ivor Lewis procedure for an intrathoracic anastomosis or three-incision McKeown procedure for a cervical anastomosis are adequate surgical techniques for resection [1]. These procedures allow the resection of the tumor-bearing esophagus, mediastinal and perigastric lymph node dissection, and reconstruction of the intestinal passage that is routinely performed as gastric pull-up. Nevertheless, abdomino-thoracic esophagectomies are high-risk surgical procedures, facing high rates of postoperative morbidity and mortality, even in high-volume surgical centers [2,3,4,5,6,7,8,9,10]. Apart from anastomotic complications, high rates of postoperative respiratory complications, especially pneumonia, are reported after abdomino-thoracic esophagectomy that cause a high postoperative morbidity, a frequent need for critical care, and poor postoperative short-term as well as unsatisfactory long-term oncological outcomes [2,4,5,6,7,8,9,11,12,13,14,15]. Multiple factors were proposed to contribute to the pathogenesis of postoperative pulmonary complications like perioperative atelectasis after single-lung ventilation, post-thoracotomy pain affecting respiratory physiology, as well as intraoperative injury to the thoracic cavity and the lung [10,15,16,17,18,19,20].

Several minimally invasive approaches to esophagectomy were introduced into clinical practice: total minimally invasive and hybrid minimally invasive procedures including laparoscopy combined with thoracotomy or thoracoscopy combined with laparotomy [1,21,22,23,24,25,26,27,28]. If minimally invasive thoracoscopy is superior to thoracotomy in the context of esophagectomy, is still a matter of debate [23,25,27,29,30]. However, ample evidence including the prospectively conducted randomized MIRO trial reported considerably higher incidences of major pulmonary complications after open esophagectomy (OE) compared to hybrid minimally invasive laparoscopically assisted esophagectomy (LAE) [21,22,24,29,30,31,32]. This might be due to reduced abdominal pain and trauma in comparison to open surgery [15,17,18,19,20,22,31,33,34,35]. As the trauma-associated release of danger-associated molecular patterns (DAMPs) is expected to directly cause pulmonary barrier dysfunction [17,19,20], a better early postoperative lung function should be expected in patients undergoing minimally invasive surgery. This is, however, not yet systematically investigated.

Previously, we compared the pulmonary impairment in patients, who underwent Ivor Lewis abdomino-thoracic esophagectomy with right-sided thoracotomy, to patients, who underwent conventional open right-sided major pulmonary resection [5]. Herein, the esophagectomy group included both, OE and LAE [5]. Although the thoracic part of the surgical procedure is largely similar among both groups, the pneumonia rate after esophagectomy amounted to almost 40% and was considerably higher than after pulmonary resection [5]. We further analyzed the time course of the Horovitz index (i.e., PaO_2_/FiO_2_ ratio (P/FR)), which is a highly sensitive indicator of any kind of respiratory impairment [5,36]. As expected, the first intraoperative P/FR was worse in the pulmonary resection group. However, already at postoperative day (POD) 0, the situation reversed and esophagectomy patients suffered more frequently from a reduced P/FR than patients after pulmonary resection [5]. This striking result led us to the conclusion that neither thoracotomy with injury to the thoracic cavity and to the lung during surgery nor single lung ventilation are the main culprits for the high rate of respiratory impairments and complications after abdomino-thoracic esophagectomies [5]. We hypothesized that partial or complete vagal denervation of the lung, bronchi, and pulmonary vasculature, that is typical for oncologic esophagectomies, causes a dysbalance of the autonomic nervous system innervating pulmonary structures and eventually impairs gas exchange [5]. Another possible explanation is the abdominal part of the esophagectomy that is associated with additional surgical trauma and more pain [15,17,18,19,20,22,31,33,34,35].

We conducted this retrospective single-center analysis to compare the pulmonary outcome of patients after a conventional OE to patients, who underwent a hybrid minimally invasive LAE. In contrast to the above mentioned studies from other authors [21,22,24,29,30,31,32], we included the highly sensitive P/FR that enables us to investigate pulmonary impairment in the early postoperative phase [5]. We critically scrutinize the common hypothesis that the minimally invasive LAE causes less pulmonary damage compared to OE because of reduced intraoperative trauma [17,18,19,20,35].

## 2. Material and Methods

### 2.1. Patients

This retrospective single-center cohort study was performed in accordance to the latest version of the Declaration of Helsinki and was approved by the local ethics committee of University of Giessen (approval No. 214/15 and 253/16). The data are collected, the manuscript is written and submitted in accordance to the COPE guidelines. All patients were treated according to the institutional standard-of-care.

From 01/2007 to 12/2017, all consecutive patients who underwent the two-staged, abomino-thoracic (Ivor Lewis) or three-incision, abdomino-thoraco-cervical (McKeown) esophagectomy for cancer were retrospectively evaluated. All patients underwent right-sided, anterolateral thoracotomy for the thoracic part and laparotomy (OE group) or laparoscopy (LAE group) for the abdominal part of the esophagectomy procedure. To focus on abdomino-thoracic procedures for esophagectomy, patients were excluded from the study, who underwent a transhiatal surgical approach to the lower thoracic esophagus. To generate homogenous cohorts of oncologic abdomino-thoracic esophagectomies, patients were excluded from the study, who underwent re-do surgery for recurrent esophageal carcinoma (*n* = 2) or multivisceral abdominal surgery with pancreatic resection for locally advanced carcinoma of the esophago-gastric junction (*n* = 2) as well as abdomino-thoracic esophagectomy for benign disease or esophageal perforation (*n* = 3), abdomino-cervical esophagectomy without a trans-thoracic part of the operation (*n* = 2), and cervical esophagectomy for carcinoma of the larynx or hypopharynx (*n* = 2). Patients were divided into two groups: patients who underwent either conventional open surgery for abdomino-thoracic esophagectomy (OE) or a hybrid minimally invasive approach with laparoscopic surgery for the abdominal part of esophagectomy (LAE). Patients who underwent conversion from the initially intended laparoscopic to conventional open abdominal surgery were assigned to the OE group. Overall, 10 patients underwent conversion from the laparoscopic approach to OE: *n* = 2 due to bleeding, *n* = 5 due to dense adhesions, *n* = 3 due to technical considerations.

Patient data were analyzed retrospectively from the prospectively maintained institutional database during the first ten postoperative days (POD 0–10) regarding the P/FR, a key parameter to evaluate perioperative pulmonary function. P/FR values were available for patients staying on intensive care unit (ICU). An important general clinical criterion for discharge of patients from the ICU to a normal ward after major surgery is adequate respiratory function. Consequently, discharge from ICU was interpreted as absence of respiratory impairment (or failure). For patients on the normal ward a “normal oxygenation” was anticipated and a P/FR ≥ 300 mm Hg was assumed. As described previously [36], the P/FR was calculated as the ratio of the arterial pressure of oxygen (PaO_2_) and the fraction of inspired oxygen (FiO_2_) (PaO_2_/FiO_2_) at the beginning of mechanical ventilation (in most cases under double-lung ventilation), at the end of surgery, upon arrival at the ICU (POD 0), and on POD 1–10 [5]. If the PaO_2_ and FiO_2_ were measured more than once a day, the first values of the day were used. For mechanically ventilated patients (either invasively or non-invasively) the FiO_2_ was available. For patients who were not mechanically ventilated (invasively or non-invasively), a FiO_2_ of 30% was anticipated. Postoperative re-intubations were assessed independently from re-do surgery, to focus on acute respiratory insufficiency making a re-intubation necessary. According to the *Berlin classification*, a P/FR ≤300 mm Hg is an important criterion for the clinical definition of ARDS (mild: 201–300 mm Hg, moderate: 101–200 mm Hg and severe: ≤ 100 mm Hg) [36], thus a P/FR <300 mm Hg was considered in this work to indicate respiratory impairment [5].

The rate of pneumonia during the first 30 postoperative days (POD 1–30) after esophagectomy was assessed as the second key parameter of the study. For the retrospective assessment of postoperative pneumonia the “Revised Uniform Pneumonia Score,” introduced and validated for patients after esophagectomy by Weijs et al. was used [37]. The postoperative day of retrospective pneumonia diagnosis was recorded for two-group comparison as well as cumulative incidence calculation. As described previously, although resulting in slightly higher pneumonia rates compared to the current literature, the scoring system was minimally modified according to current “International Guidelines for the Management of Severe Sepsis and Septic Shock 2012,” and we decided to use a body temperature ≥ 38.0 °C or ≤ 36.0 °C, respectively, as the threshold for pneumonia scoring in our study [5,37,38].

Postoperative pneumonia and perioperative P/FR were investigated as the main outcome parameters of the present study. The following secondary outcome parameters were assessed: characteristics of patients, tumors as well as surgical procedures, perioperative markers of inflammation (leukocyte count and C-reactive protein (CRP) in peripheral blood obtained from routine laboratory examinations), postoperative pulmonary outcome parameters (duration of mechanical ventilation, rates of mechanical ventilation during POD 0–10, reintubation and tracheotomy rates), and major general patient outcome parameters (duration of stay at the ICU and duration of total in-hospital stay, rates of anastomotic complications, re-do surgery, and overall mortality). Duration of postoperative stay data were stratified into initial postoperative stay at the ICU, cumulative postoperative stay at the ICU, and total postoperative in-hospital stay. The “cumulative postoperative stay at the ICU” variable was calculated for patients who underwent repeated referrals to the ICU during the postoperative in-hospital stay.

### 2.2. Surgery

Abdomino-thoracic esophagectomy for esophageal malignoma or cancer of the esophago-gastric junction is a widely used standard surgical procedure. The institutional technique was described previously [5]. Briefly, a median laparotomy from the xiphoid to the umbilicus was performed for the open abdominal part of Ivor Lewis as well as McKeown esophagectomy. For the hybrid minimally invasive approach we used a minilaparotomy for the camera access and a four-port technique for laparoscopic surgery. Basically, the technique of gastric mobilization and tube building as well as perigastric, abdominal lymph node dissection was the same for OE and LAE. All abdomino-thoracic esophagectomy patients underwent a right-sided, anterolateral thoracotomy in the 4 th or 5 th intercostal space as the standard access to the thoracic cavity for the thoracic part of the operation. In line with the local clinical standard, abdomino-thoracic esophagectomy was performed as a two-stage procedure in one surgical intervention. In most cases the thoracic part followed the abdominal part of surgery. Reasons for a thorax-first approach were the determination of local resectability in cases of suspected infiltration of neighboring thoracic organs. The subtotal esophagectomy and reconstruction the esophago-gastric continuity was completed trans-thoracally by gastric pull-up. Only in one case included into this study the continuity was restored by colonic interposition following esophago-gastrectomy. Two-field lymph node dissection was performed as the standard procedure during oncologic esophagectomy, except in cases of abdomino-thoraco-cervical esophagectomies with cervical anastomoses, lymph node dissection was completed as a three-field procedure following international recommendations [1]. The decision for OE or LAE was based on the surgeon’s preferences and tumor stages. OE was preferred, especially when perigastric lymph node metastases were suspected and in cases of locally advanced tumors of the esophago-gastric junction. Gastric tube was stapled from the small gastric curvature and the anastomosis was done using circular stapler devices. The duration of the thoracic part of the surgical procedure was estimated by the duration of single-lung ventilation or the time of the thoracic incision, respectively.

In the postoperative phase, patients were treated by principles of a “fast track” protocol with extubation as soon as possible, early enteral nutrition and mobilization at the earliest convenience [39]. Usually, patients were monitored at the ICU after abdomino-thoracic esophagectomy for at least until POD 1 and—if cardiac and respiratory functions were stable—discharged from the ICU. 

### 2.3. Statistical Analysis

Statistical analyses were performed using GraphPad Prism (Version 5.00 for Windows, GraphPad Software, San Diego, CA, USA, www.graphpad.com). For descriptive statistics, data of both groups were analyzed using Fisher’s exact or Pearson’s X^2^ test for categorical data in cross-tabulation. Two group comparisons of continuous variables were performed by Mann-Whitney-U test. Patients who died were censored from analysis upon the day of death (indicated in the tables).

To determine a statistical dependence, the duration of surgery (total, thoracic part, abdominal part) was correlated with the last intraoperatively assessed P/FR as well as P/FR on POD 0–3 by Spearman’s Rho rank correlation. If P/FR were not available on POD 1–3 due to the reasons mentioned before, a P/FR = 300 mm Hg was assumed for the calculation. Results are given as the Spearman’s rank correlation coefficient (r_sp_).

Cumulative incidences of postoperative pneumonia during POD 1–30 and postoperative respiratory impairment (defined by a P/FR < 300 mm Hg) during POD 1–10 of patients after OE or LAE were calculated by Kaplan-Meier estimation. Log-rank test was used to compare Kaplan-Meier curves. Patients who were discharged, died, or underwent re-do surgery were censored from the analysis of cumulative incidences. Censored data are indicated in the figures by vertical ticks. A P/FR <300 mm Hg on POD 0 (arrival on ICU) was not judged as postoperative event.

Data are given in tables as medians and minimum to maximum ranges for continuous variables as well as *n* (%) for categorical variables; *p*-values ≤ 0.05 were considered to indicate statistical significance.

Post hoc power calculations and sensitivity analysis was performed using the freely available software G*Power 3.1 [40].

## 3. Results

### 3.1. Patients

In total, 143 patients, who underwent abdomino-thoracic surgery for esophageal malignancy, were included in this study. Among them, 105 patients underwent OE and 38 patients LAE. Except for the rate of arterial hypertension, which was more prevalent in the OE group (*p* = 0.02), preoperative patient characteristics resembled in both groups regarding body mass index, American society of Anesthesiologist’s classification of physical health (ASA) score as well as chronic cardiac and pulmonary diseases. Even tumor characteristics and the rate of induction therapies (chemo- or radio-chemo-therapy) embedded in multimodal treatment strategies were similar between both groups (Table 1).

### 3.2. Surgical Procedure

During surgery, the intestinal continuity was restored by a cervical anastomosis in 12 patients of the OE group and in one patient of the LAE group (*p* = 0.19). Furthermore, 36 patients from the OE group and 10 patients from the LAE group underwent an extended surgical procedure (*p* = 0.42), while one patient of the OE group underwent esophagogastrectomy with colonic interposition. The surgical procedure was most frequently extended to minor lung or liver resections as well as to cholecystectomy or omentectomy (Table 2).

The total duration of surgery (*p* = 0.07), the duration of the thoracic part of esophagectomy and hence, also the duration of single-lung ventilation was longer in LAE patients (*p* = 0.004, Table 2). Of note, the remaining time needed for the abdominal part of the procedure and for intraoperative re-positioning did not differ between both groups (OE: 161 (82–408) min vs. LAE: 170 (90–276) min, *p* = 0.66), indicating that the laparoscopic approach for the abdominal part of surgery was not more time consuming than the open abdominal approach.

### 3.3. Inflammation

Leukocyte counts and C-reactive protein (CRP) levels in peripheral blood showed a peak on POD 2 after esophagectomy. While perioperative leukocyte counts did not differ among both groups, perioperative CRP levels, were slightly higher on POD 0 (*p* = 0.05) and tended to be higher on POD 1 (*p* = 0.07) in patients, who underwent OE compared to the LAE group (Table 3).

### 3.4. General Outcome

No statistically significant differences were observed between both groups regarding the initial and cumulative postoperative stays on the ICU as well as the total postoperative duration of hospitalization. Patients, who died during hospitalization, were excluded from calculation of postoperative lengths of stays. No differences were observed concerning the rates of perioperative catecholamine administration, re-do surgery (during POD 1–30), anastomotic complications requiring an intervention, and 30-day or total in-hospital mortality (Table 4).

### 3.5. Pulmonary Outcome

Cumulative incidences of postoperative pneumonia between POD 1–30, which was adjusted to re-do surgery, mortality and discharge from hospital, were significantly different among the OE and LAE group (*p* = 0.01, Figure 1). The overall rate of postoperative pneumonia, regardless of re-do surgery, death or discharge, was 45.7% and 26.3% among patients from the OE and LAE group, respectively (*p* = 0.05, Table 5). To evaluate this result, we conducted a post hoc power analysis (by G*Power 3.1 [40]) based on our data. With an alpha error probability of 5%, the analysis yielded a moderate power of 0.61 and an actual alpha value of 0.032. Vice versa, a sample size of 159 respectively 57 patients (therewith maintaining the reported group proportion) would have been needed to achieve a power of 0.8 (with alpha error probability of 0.05), assuming the reported effect size. A subsequent sensitivity analysis based on sample size and alpha error probability further showed that a proportion of 51% pneumonia would have increased the actual power to 0.8.

Only rarely, a P/FR < 300 mm Hg was measured at the beginning of surgery. However, the majority of patients from both groups had a reduced P/FR (<300 mm Hg) during POD 1–10 after esophagectomy (74.3% and 84.2% of the patients from the OE and LAE group, respectively, *p* = 0.27). This indicates a high rate of respiratory impairment after surgery, with a peak on POD 2 in both groups (Table 5). Consequently, no differences were observed in the cumulative incidences of reduced P/FR during POD 1–10 between the OE and LAE group (*p* = 0.54, Figure 2). Not surprisingly, post hoc analysis computes a low power of 0.27 for this comparison.

No differences between both groups have been observed in the initial extubation success and in the direct comparison of rates of reduced P/F ratios during the early postoperative phase (POD 0–2). By contrast, a reduced P/FR was more frequently observed on POD 3, 5, and 10 (*p* ≤ 0.05) in patients from the OE group and this respiratory impairment tended to persist on POD 7 and 9 (*p* ≤ 0.1, Table 5). This may reflect the higher rate of invasive mechanical ventilation on POD 7–10 (*p* < 0.05) as well as higher rate of acute respiratory insufficiency necessitating re-intubation independent of re-do surgery (*p* = 0.05). Accordingly, a longer duration of cumulative perioperative mechanical ventilation (*p* = 0.05) was seen in the OE group (Table 4 and Table 5).

Interestingly, although the duration of surgery tended to be longer in LAE patients, we found negative correlations between the total duration of surgery and P/F ratios on POD 2 and 3 in the OE group (r_Sp_ = −0.194, *p* = 0.05 and r_Sp_ = −0.267, *p* = 0.006, respectively). This also holds true for the correlation of the duration of the abdominal part of surgery and P/FR in OE patients on POD 3 (r_Sp_ = −0.242, *p* = 0.01, Table 6).

## 4. Discussion

Respiratory complications, mainly pneumonia, are a severe problem after esophageal surgery and at least in part responsible for the frequent need for postoperative critical care as well as for the high morbidity and mortality [2,4,5,6,7,8,9,11,12,13,14,15,16,17,18,19,20]. A better understanding of the presumably multifactorial pathogenesis of postoperative pulmonary impairment [15] is mandatory for the development of new preventive and therapeutic strategies that are urgently needed.

The present study that compares OE to LAE is a relatively small, retrospective single-center analysis with all known inherent draw-backs and limitations. Apart from the retrospective character of the study, the assumption of a FiO_2_ = 30% for the P/FR calculation of patients, who were not ventilated as well as the assumption of a P/FR = 300 mm Hg for patients on the normal ward without respiratory impairment are further limitations of this study. For patients, who did not obtain any kind of ventilation on the ICU but frequently receive nasal oxygen supply the FiO_2_ was set to 30% in our calculation. Patients with missing information regarding PaO_2_ and FiO_2_ usually stayed on the normal ward without respiratory impairment. Thus, for the calculation of r_sp_ the P/FR was set to 300 mm Hg, which corresponds to the lower limit of a normal oxygenation [36]. Hence, our data should be interpreted with utmost care, and no firm conclusions can be drawn. However, our study allows the creation of testable hypotheses. Our results suggest that the pathogenesis of pulmonary morbidity in response to esophagectomy is even more complex than expected.

Our study suggests that two distinct phases of postoperative pulmonary impairment can be discerned: (i) during POD 0–2, the rate of patients with a P/FR < 300 mm Hg is high, irrespective of OE or LAE. Rates of reduced P/F ratios in the early postoperative phase (POD 0–2) were not caused by pneumonia or anastomotic complications [5]. (ii) Later on (POD 3, 5, 7, 9, and 10) a higher prevalence of a P/FR < 300 mm Hg is seen in OE patients, which presumably reflects the higher incidence of pneumonia in this group. This time-line is important for the treating physicians, especially regarding discharge considerations of esophagectomy patients from the intensive care unit to a normal ward.

We confirm the result of numerous studies including the recent prospective randomized MIRO trial, that the cumulative incidence of major postoperative pulmonary complications, especially pneumonia is significantly higher after OE compared to LAE [21,22,24,29,30,34,41]. In comparison to these studies, we observed slightly higher rates of postoperative pneumonia, presumably because we used the minorly modified “Revised Uniform Pneumonia Score” by Weijs et al. for the retrospective assessment of pneumonia [37]. The threshold of the body temperature was adjusted according to the current “International Guidelines for the Management of Severe Sepsis and Septic Shock 2012” [38], which in turn lead to a higher pneumonia rate than that reported in the recent literature.

However, our observation that the P/FR, a sensitive marker of pulmonary dysfunction, does not differ among both patient groups early after surgery (POD 0–2), argues against the common interpretation, that the higher incidence of pneumonia in the OE group was directly due to an increased surgical trauma, to more postoperative pain or to a higher incidence of surgery-induced basal lung damage [10,15,16,17,18,19,20,21]. The effects of these factors would be expected to be visible early after surgery, which was obviously not the case in our study. We hypothesize that the rate of early impairment of pulmonary function after esophagectomy, that is surprisingly similar in the LAE and in the OE group, is not importantly influenced by the extent of the trauma and even pain caused by the laparotomy during the abdominal part or injury of the thoracic cavity by the thoracic part of the esophagectomy procedure.

We reported before, that especially the early postoperative pulmonary impairment of patients undergoing Ivor Lewis esophagectomy is more pronounced than that of patients undergoing lung resection [5]. A possible explanation is that the thoracic part of esophagectomy results in pulmonary dysfunction due to an imbalance in the autonomous nervous system caused by vagal denervation of the lung [5,42]. This imbalance might cause an early postoperative situation resembling neurogenic pulmonary edema [5,43,44]. Indeed, an extended damage of vagal nerve fibers to pulmonary structures due to mediastinal lymph node dissection and the transthoracic esophagectomy itself as well as the technical feasibility of vagus-sparing techniques during abdomino-thoracic esophagectomy have been previously described [42,45,46,47,48,49]. However, vagus-sparing techniques bare the much feared risks of a lower oncological radicality [42,45,50]. Nevertheless, in some small case series a reduced prevalence of complications driven by vagal damage as well as pulmonary complications after esophagectomy was reported if care was taken for the vagal nerve during surgery [42,45,46,47,48,49]. If damage to the vagal nerve is the major threat for the lung in the context of esophagectomy, it is conceivable that the early postoperative P/F ratios are similar in the OE and in the LAE group. Vagus-sparing techniques should be carefully evaluated in prospective multi-center studies, because they might reduce respiratory impairment and pneumonia rates after LAE below those observed after pulmonary resection.

Nevertheless, we still need to explain, why the incidence of postoperative pneumonia beyond POD 3 is much higher in the OE group compared to the LAE group. As the cumulative incidence of pneumonia in the LAE group is close to that of patients, who underwent pulmonary resection [5], the damage associated with laparotomy but not with laparoscopy seems, at least in part, to be responsible for the increased incidence of pneumonia after esophagectomy compared to pulmonary resection. As discussed above, a direct effect of a trauma-induced release of DAMPs is unlikely, although the elevated CRP levels at POD 0 suggest that more DAMPs were released in OE patients. However, the negative correlation of the duration of surgery in the OE group and the P/FR on POD 3 would be in favor of the idea that lager amounts of DAMPs released during prolonged major abdominal surgery increase the risk to acquire pneumonia [51,52,53]. We speculate that a more severe initial DAMP-induced systemic inflammation might have impaired host defense against pneumonia-associated pathogens. If this holds true, LAE should be favored whenever possible. Similar conclusions can be drawn from the recent study by Babic et al. [54]. Here the open esophagectomy approach resulted in higher levels of inflammatory markers (CRP and blood leukocytes) during the first two postoperative days compared to minimally invasive approaches including the hybrid one. In the their analysis Babic et al. evaluated high early postoperative CRP values as positive predictive factors for the development of postoperative overall as well as pulmonary complications [54]. Another variable, which was significantly different between both the OE and LAE group in the present study, was intraoperative blood loss. The most probable explanations for respiratory impairment due to an extensive intraoperative blood loss are volume overload or transfusion-associated lung injury, but the intraoperative rate of blood product transfusion and respiratory impairment did not correlate. However, blood loss negatively correlated with P/FR on POD 3 in the OE group (see Table 6). In our study, POD 3 seems to be the day with the highest risk for developing lung injury following OE and the immunological impact of intraoperative blood loss and perioperative transfusion should be further investigated. Another trivial explanation for the higher incidence of pneumonia in the OE group might be respiratory impairment due to larger painful operation scars, although our center attaches much importance to an appropriate pain management. Besides that, Klevebro et al. evaluated higher pneumonia and pulmonary complication rates in patients with cardiorespiratory co-morbidities, especially in patients with lung diseases after neoadjuvant therapies, in their multicentric analysis of 1590 esophagectomies [14]. Although one could suspect, that because of the retrospective and non-randomized nature of the current study, the OE procedure was preferentially performed in patients with preoperatively suspected more advanced tumor stages and a higher general morbidity, our data speak against this theory: both study groups are widely balanced concerning the (local) tumor stages and showed only differences in pre-existing arterial hypertension—not in chronic lung or cardiac diseases. Differences in irradiation-induced lung pathology, that might foster postoperative pneumonia [2,55,56], can also be excluded in our study, as the rate of neoadjuvant therapy did not differ between both groups of the present study. Although, other factors might contribute to the incidences of pneumonia as well as reduced P/FR after abdomino-thoracic esophagectomy, we decided not to perform a multivariable analysis in order to focus on the results of two-group comparisons, incidence analyses and rank correlations and to avoid statistical overstatement in the present work.

## 5. Conclusions

In summary, our study suggests that an early phase of postoperative pulmonary impairment that starts immediately after surgery is inherent to the open thoracic part of the esophagectomy procedure but largely independent of the extent of abdominal trauma. Prospective randomized clinical studies are warranted to improve the pulmonary outcome of esophagectomy and to decide if a minimally invasive approach to the thoracic part, a vagus-sparing technique or both is of advantage. The slightly delayed development of pneumonia is most likely driven by the initial pulmonary damage after OE and LAE. It is less likely that the laparotomy itself is responsible for the increased postoperative pneumonia rate.

## Figures and Tables

**Figure 1 jcm-09-01896-f001:**
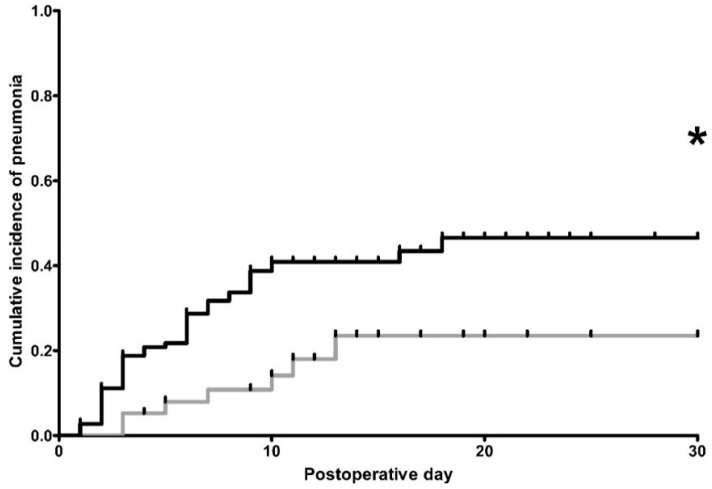
Kaplan-Meier estimation of cumulative incidences of postoperative pneumonia. Black line: conventional open abdomino-thoracic esophagectomy (OE)-group, *n* = 105 patients. Gray line: hybrid minimally invasive, laparoscopically assisted abdomino-thoracic esophagectomy (LAE)-group, *n* = 38 patients. Patients, who were discharged, died, or suffered from re-do (revision) surgery were censored from the analysis of cumulative incidences since the day of the event. Censored data are indicated in the figures by vertical ticks. * indicates differences in the cumulative incidence of postoperative pneumonia between both groups at postoperative day 30 (*p* = 0.01).

**Figure 2 jcm-09-01896-f002:**
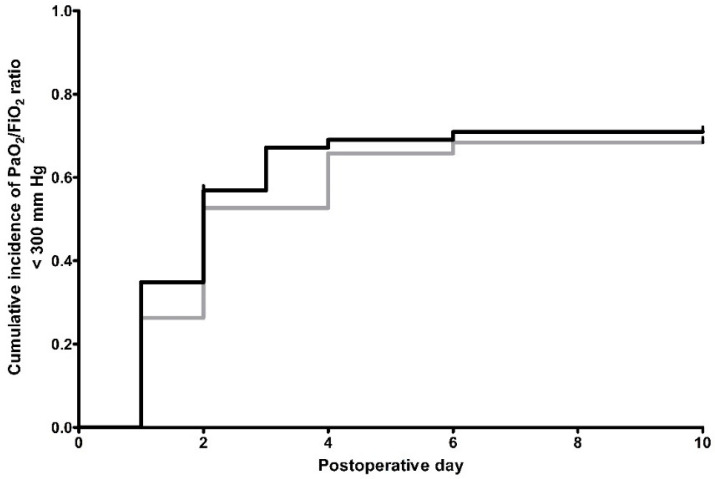
Kaplan-Meier estimation of PaO_2_/FiO_2_ < 300 mm Hg, indicating respiratory impairment. Black line: conventional open abdomino-thoracic esophagectomy (OE)-group, *n* = 105 patients. Gray line: hybrid minimally invasive, laparoscopically assisted abdomino-thoracic esophagectomy (LAE)-group, *n* = 38 patients. Patients, who were discharged, died, or suffered from re-do (revision) surgery were censored from the analysis of cumulative incidences since the day of the event. Censored data are indicated in the figures by vertical ticks. An PaO_2_/FiO_2_ of <300 mm Hg on postoperative day 0 (arrival on intensive care unit) was not judged as postoperative event.

**Table 1 jcm-09-01896-t001:** Patient characteristics.

Variables	Conventional Open Esophagectomy (*n* = 105)	Laparoscopically Assisted Esophagectomy (*n* = 38)	*p*-Value
Male gender (*n*)	86 (81.9%)	33 (86.8%)	0.62
Age (years)	64 (40–86)	62.5 (42–77)	0.34
BMI (kg/m^2^)	24.5 (15.6–41.3)	24.1 (16.2–31.7)	0.35
ASA (median)	3 (1–4)	2 (1–3)	0.43
1 (*n*)	5	2	
2 (*n*)	44	18	
3 (*n*)	52	18	
4 (*n*)	4	0	
History of malignancy (*n*)	19 (18.1%)	5 (13.2%)	0.62
Arterial hypertension (*n*)	67 (63.8%)	16 (42.1%)	0.02
Coronary artery disease (*n*)	18 (17.1%)	7 (18.4%)	0.81
Chronic lung disease (*n*)	19 (18.1%)	7 (18.4%)	1
Chronic kidney failure (*n*)	7 (6.7%)	3 (7.9%)	0.73
Induction therapy (*n*)	57 (54.3%)	22 (51.9%)	0.85
Chemo	22	9	
Radio–Chemo	35	13	
Indication (*n*)			0.65
Malignancy	100%	100%	
Adenocarcinoma	62 (59.0%)	24 (63.2%)	
Squamous cell	40 (38.1%)	12 (31.6%)	
carcinoma			
Other	3 (2.9%) *	2 (5.3%) ^#^	
Pathological tumor stage (*n*) ^§^			
T 0	13 (12.6%)	7 (19.4%)	0.71
T 1	13 (12.6%)	3 (8.3%)	
T 2	30 (29.1%)	11 (30.6%)	
T 3	46 (44.7%)	14 (38.9%)	
T 4	1 (1.0%)	1 (2.8%)	
N 0	52 (50.5%)	18 (50%)	1
N +	51 (49.5%)	18 (50%)	
M + ^$^	11 (10.7%)	1 (2.3%)	0.3

* Including neuroendocrine carcinoma and gastrointestinal stromal tumor (each *n* = 1) as well as one salvage esophagectomy for suspected malignant stenosis after primary radio-chemo-therapy 14 years before. # Including sarcoma and mucosal melanoma. § including adenocarcinoma and squamous cell carcinoma, regarding the current UICC-classification. $ intraoperatively detected oligo-metastatic disease in all patients.

**Table 2 jcm-09-01896-t002:** Procedure characteristics.

Variables	Conventional Open Esophagectomy (*n* = 105)	Laparoscopically Assisted Esophagectomy (*n* = 38)	*p*-Value
Main procedure			0.19
Thoracic anastomosis	93 (88.6%)	37 (97.4%)	
Cervical anastomosis	12 (11.4%)	1 (2.6%)	
Lymph node dissection	100%	100%	1
Relevant abdomino/thoracic extended procedures (additional to main procedure) (*n*)	*n* patients = 35 (33.3%)	*n* patients = 10 (26.3%)	0.54
	esophagogastrectomy: 1 (1.0%) *	esophagogastrectomy: 0	1
	Major lung resection: 1 (1.0%)	Major lung resection: 1 (2.6%)	0.46
	Minor lung resection: 7 (6.7)	Minor lung resection: 4 (10.5%)	0.48
	Minor liver resection: 13 (12.4%)	Minor liver resection: 1 (2.6%)	0.11
	Jejunum catheter: 4 (3.8%)	Jejunum catheter: 0	0.57
	Cholecystectomy: 3 (2.9%)	Cholecystectomy: 1 (2.6%)	1
	Colon resection: 1 (1.0%)	Colon resection: 2 (5.3%)	0.17
	Appendectomy: 2 (1.9%)	Appendectomy: 0	1
	Omentectomy: 4 (3.8%)	Omentectomy: 1 (2.6%)	1
	Left adrenalectomy: 2 (1.9%)	Left adrenalectomy: 0	1
	Other minor resections: 3 (2.9%)	Other minor resections: 2 (5.3%)	0.61
Duration of the thoracic part of Ivor Lewis procedure (min) ^#^	118 (45–304) ^§^	146.5 (86–423)	0.004
Total duration of surgery (min)	288 (177–537)	315 (190–635)	0.07
IO Blood loss (mL)	600 (100–4800)	432.5 (50–2500)	0.01
IO transfusion (*n* patients) ^&^	28 (26.7%)	7 (18.4%)	0.38
Peridural anesthesia (*n*)	76 (72.4%)	30 (78.9%)	0.52

* One patient underwent esophago-gastrectomy with colon interposition. # Duration of single-lung ventilation or total duration of the thoracic part of esophagectomy, depending on retrospective availability of data. § Not available retrospectively in 2 patients. & Including packed red blood cells and fresh frozen plasma. IO = intraoperative.

**Table 3 jcm-09-01896-t003:** Perioperative leukocyte counts and C-reactive protein levels.

Variables	Conventional Open Esophagectomy (*n* = 105)		Laparoscopically Assisted Esophagectomy (*n* = 38)		*p*-Value
Leukocytes (giga/L)		missing		missing	
		values		values	
pre–operatively	6.6 (2.0–16.4)	–	6.7 (3.6–15.5)	–	0.83
POD 0 (on arrival at ICU)	9.3 (3.0–29.6)	–	7.8 (4.3–28.9)	1	0.54
POD 1	10.5 (4.0–23.6)	–	10.2 (6.2–19.1)	1	0.72
POD 2	11.4 (1.8–21.9)	–	11.4 (3.7–24.6)	–	0.85
POD 3	9.5 (1.9–34.1)	4	9.0 (5.5–21.5)	4	0.57
POD 4	8.1 (2.7–106.0)	12 *	7.9 (3.9–17.9)	6	0.37
POD 5	7.8 (3.4–22.0)	20 *	7.8 (3.9–16.3)	9	0.46
POD 6	9.1 (3.4–26.1)	24 ^#^	9.0 (4.8–17.6)	14	0.4
POD 7	9.9 (3.0–29.0)	26 ^#^	9.3 (5.4–23.0)	13	0.72
POD 8	11.0 (4.1–33.3)	32 ^#^	10.0 (5.9–22.5)	17	0.41
POD 9	12.3 (4.5–49.7)	29 ^#^	10.4 (5.3–30.2)	16	0.37
POD 10	12.6 (4.7–38.7)	36 ^§ &^	10.4 (4.2–36.7)	18 ^€^	0.29
at discharge	8.1 (3.1–19.0)	13 ^$^	6.9 (4.2–15.7)	2 ^#^	0.07
C–reactive protein (mg/L)		missing values		missing values	
pre–operatively	3.8 (0.5–159.1)	–	2.4 (0.5–124.4	–	0.53
POD 0 (on arrival at ICU)	6.8 (0–256.0)	5	3.1 (0.5–76.6)	2	0.05
POD 1	94.4 (31.6–226.2)	–	78.0 (31.2–205.2)	1	0.07
POD 2	199.2 (55.3–359.4)	–	192.9 (97.7–329.3)	–	0.6
POD 3	185.7 (26.3–403.9)	4	161.3 (69.2–359.5)	4	0.26
POD 4	159.8 (30.1–410.0)	12 *	129.0 (64.0–391.0)	6	0.15
POD 5	136.4 (25.4–539.1)	20 *	135.6 (32.7–287.7)	9	0.5
POD 6	129.8 (14.1–423.2)	23 ^#^	119.7 (26.6–281.4)	14	0.66
POD 7	123.1 (8.3–445.1)	26 ^#^	121.9 (27.4–333.4)	13	0.74
POD 8	144.5 (6.0–491.9)	33 ^#^	131.5 (30.4–361.7)	17	0.72
POD 9	141.9 (5.6–446.9)	30 ^#^	138.0 (27.1–283.6)	18	0.97
POD 10	153.4 (8.1–393.9)	36 ^§ &^	109.3 (4.9–302.2)	18 ^€^	0.19
at discharge	32.0 (1.4–145.1)	13 ^$^	34.4 (4.9–144.9)	2 ^#^	0.31

* including 1 death. # including 2 deaths. § including 3 deaths. & including 2 patients who were discharged on postoperative day 9. € including 4 patients who were discharged on postoperative day 9. $ including 13 deaths.

**Table 4 jcm-09-01896-t004:** Perioperative results.

Variables	Conventional Open Esophagectomy (*n* = 105)	Laparoscopically Assisted Esophagectomy (*n* = 38)	*p*-Value
PO hospital stay Total (d) *	18 (9–141)	14.5 (9–75)	0.14
Initial PO stay at the ICU (d) *	5 (1–76)	4 (1–35)	0.15
Return to ICU (*n* patients)	17 (16.2%)	6 (15.8%)	1
Cumulative PO stay at the ICU (d) *	5.5 (1–84)	5 (1–35)	0.16
Cumulative perioperative mechanical ventilation (h)	17.3 (4.8–2280)	12.6 (5.3–26.3)	0.05
Rate of invasive PO ventilation (*n*) ^#^			
POD 0 (on arrival at ICU)	83 (79.0%)	32 (84.2%)	0.64
POD 1	46 (43.8%)	17 (44.7%)	1
POD 2	12 (11.4%)	2 (5.3%)	0.35
POD 3	18 (17.1%)	1 (2.6%)	0.03
POD 4	20 (19.2%) ^§^	3 (7.9%)	0.13
POD 5	21 (20.2%) ^§^	3 (7.9%)	0.13
POD 6	21 (20.4%) ^$^	3 (7.9%)	0.13
POD 7	26 (25.2%) ^$^	1 (2.6%)	0.001
POD 8	23 (22.3%) ^$^	2 (5.3%)	0.02
POD 9	20 (19.4%) ^$^	1 (2.6%)	0.01
POD 10	20 (19.6%) ^&^	2 (5.3%)	0.04
PO catecholamine administration (*n* patients) ^# €^			
POD 0	48 (45.7%)	14 (36.8%)	0.45
POD 1	46 (43.8%)	10 (26.3%)	0.08
POD 2	32 (30.5%)	9 (23.7%)	0.53
POD 3	29 (27.6%)	6 (15.8%)	0.19
POD 4	22 (21.2%) ^§^	5 (13.2%)	0.34
POD 5	17 (16.3%) ^§^	4 (10.5%)	0.59
POD 6	16 (15.5%) ^$^	4 (10.5%)	0.59
POD 7	20 (19.4%) ^$^	3 (7.9%)	0.13
POD 8	16 (15.5%) ^$^	2 (5.3%)	0.16
POD 9	16 (15.5%) ^$^	2 (5.3%)	0.16
POD 10	19 (18.6%) ^&^	3 (7.9%)	0.19
Re–do (revision) surgery during POD 1–30	17 (16.2%)	3 (7.9%)	0.28
Anastomotic complications (*n* patients) ^¶^	20 (19.0%)	5 (13.2%)	0.47
PO in–hospital mortality (*n*) ^¥^	13 (12.4%)	2 (5.3%)	0.36

* Patients who suffered from in-hospital mortality were excluded from analysis of postoperative ICU and total hospital stays. # Patients who died during POD 0–10 (*n* = 4, two in each group) were excluded from further analysis after their death: § including 1 death. $ including 2 deaths. & including 3 deaths. € including arterenol and/or dobutamine. ¶ anastomotic complications, i.e., insufficiency and/or gastric tube necrosis requiring therapy (i.e., stent, endo-vacuum therapy or re-do surgery). ¥ indicating the in-hospital mortality, even exceeding the 30-day mortality. The overall 30-day mortality rate was 7.0% (*n* = 10 patients). PO = postoperative. ICU = intensive care unit, includes medium care unit. POD = postoperative day.

**Table 5 jcm-09-01896-t005:** Pulmonary outcome.

Variables	Conventional Open Esophagectomy (*n* = 105)	Laparoscopically–Assisted Esophagectomy (*n* = 38)	*p*-Value
Pneumonia (*n* patients) *	48 (45.7%)	10 (26.3%)	0.05
Pneumonia diagnosis on POD	5 (1–25)	8.5 (3–14)	0.03
Tracheotomy (*n* patients)	19 (18.1%)	2 (5.3%)	0.06
Initial extubation during first 12 h postoperatively (*n* patients)	82 (78.1%)	30 (78.9%)	1
Re–intubation (*n* patients) ^#^	31 (29.5%)	5 (13.2%)	0.05
Perioperative PaO_2_/FiO_2_ < 300 mm Hg (*n* patients) ^§ $^			
Overall during POD 1–10	78 (74.3%)	32 (84.2%)	0.27
First intraoperative	25 (24.0%) ^&^	5 (13.2%)	0.25
Last intraoperative	61 (58.7%) ^&^	25 (65.8%)	0.56
POD 0 (on arrival at ICU)	39 (37.1%)	10 (26.3%)	0.32
POD 1	38 (36.2%)	10 (26.3%)	0.32
POD 2	50 (47.6%)	14 (36.8%)	0.34
POD 3	44 (41.9%)	7 (18.4%)	0.01
POD 4	30 (28.8%) ^ß^	8 (21.1%)	0.4
POD 5	29 (27.9%) ^ß^	4 (10.5%)	0.04
POD 6	28 (27.2%) ^€^	7 (18.4%)	0.38
POD 7	27 (26.2%) ^€^	4 (10.5%)	0.07
POD 8	26 (25.2%) ^€^	7 (18.4%)	0.5
POD 9	24 (23.3%) ^€^	4 (10.5%)	0.1
POD 10	23 (22.5%) ^¶^	3 (7.9%)	0.05

* Overall pneumonia rate, irrespectively from re-do surgery. Pneumonia was retrospectively assessed by the “Revised Uniform Pneumonia score” by Weijs et al. using the body temperature ≥38.0 °C or ≤36.0 °C [37] during postoperative days 1–30. # re-intubation because of acute respiratory insufficiency during the hospital stay, independently from re-do (revision) surgery making re-intubation necessary. § Irrespectively from re-do surgery. $ Patients who died during POD 0–10 (*n* = 3 in the open esophagectomy group) were excluded from further analysis after their death. & Not available retrospectively in 1 patient. ß excluding one death, € excluding 2 deaths, ¶ excluding 3 deaths. POD = postoperative day. ICU = intensive care unit.

**Table 6 jcm-09-01896-t006:** Results of Spearman´s Rho rank correlation.

	PaO_2_/FiO_2_ Ratio				
	Last IO (mm Hg)	POD 0 (mm Hg)	POD 1 (mm Hg)	POD 2 (mm Hg)	POD 3 (mm Hg)
Total duration of surgery (min)					
OE-group					
Correlation coefficient	0.042	−0.017	−0.146	−0.194	−0.267
*p*-value (two-sided)	0.67	0.87	0.14	0.05	0.01
Total duration of surgery (min)					
LAE-group					
Correlation coefficient	−0.079	0.116	−0.047	−0.028	−0.187
*p*-value (two-sided)	0.64	0.49	0.78	0.87	0.26
Duration of the thoracic part (min)					
OE-group					
Correlation coefficient	−0.014	−0.017	−0.136	−0.128	−0.205
*p*-value (two-sided)	0.89	0.87	0.17	0.20	0.04
Duration of the thoracic part (min)					
LAE-group					
Correlation coefficient	−0.253	0.003	−0.050	−0.115	−0.008
*p*-value (two-sided)	0.13	0.98	0.77	0.49	0.96
Duration of the abdominal part (min)					
OE-group					
Correlation coefficient	0.017	−0.035	−0.098	−0.114	−0.242
*p*-value (two-sided)	0.86	0.72	0.32	0.25	0.01
Duration of the abdominal part (min)					
LAE-group					
Correlation coefficient	0.085	0.117	0.064	0.165	−0.178
*p*-value (two-sided)	0.61	0.48	0.70	0.32	0.29
IO blood loss (mL)					
OE-group					
Correlation coefficient	0.201	0.117	−0.062	−0.076	−0.209
*p*-value (two-sided)	0.04	0.23	0.53	0.44	0.03
IO blood loss (mL)					
LAE-group					
Correlation coefficient	0.073	0.011	−0.039	−0.045	0.056
*p*-value (two-sided)	0.66	0.95	0.81	0.79	0.74
IO transfusion (*n* patients)					
OE-group					
Correlation coefficient	0.137	0.164	0.190	0.086	−0.090
*p*-value (two-sided)	0.17	0.09	0.05	0.38	0.36
IO transfusion (*n* patients)					
LAE-group					
Correlation coefficient	0.170	0.160	0.245	0.121	0.174
*p*-value (two-sided)	0.31	0.34	0.14	0.47	0.30

OE = open abdomino-thoracic esophagectomy. LAE = minimally-invasive, laparoscopically-assisted abdomino-thoracic esophagectomy. (Spearman’s rank) Correlation coefficient = r_Sp_. IO = intraoperative. POD = Postoperative day. PaO_2_/FiO_2_ ratio on postoperative day 3 correlated inversely with the duration of surgery in patients after open esophagectomy.

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
