# Peer review of "Early Respiratory Impairment and Pneumonia after Hybrid Laparoscopically Assisted Esophagectomy—A Comparison with the Open Approach"

_jcm, 2020, doi:10.3390/jcm9061896_

Round 1
Reviewer 1 Report
Title and Abstract.
- Study’s design is not reported in the title or the abstract. Authors just reported the terms “retrospective single center analysis of patients” but this does not identify a specific study type (cohort, case-control, cross-sectional,…).
- I think that summary reported in the abstract is not very clear because authors reported many p-values without incidence values used for comparisons. Moreover, the meaning of acronym rsp was not reported.
- Authors should better report the parameter used to evaluate respiratory impairment. They used Horovitz index (PF ratio) less than 300 mmHg and not the oxygenation index (OI = FiO2/PaO2 * mean airway pressure).
- In the keywords, I think the words “oxygenation index” and “acute respiratory syndrome” should be removed.
Introduction.
- Introduction is well written and I only suggest to remove the last sentence because it reports a result of the present study. I also suggest to call the Horovitz index as PF ratio or PaO2/ FiO2. This last comment is valid for the whole manuscript.
Materials and methods.
- The type of study design should be clearly reported.
- Authors reported “For patients who were not mechanically ventilated (invasively or non-invasively), a FiO2 of 30% was anticipated.” Why did the authors decide to apply this assumption? Why the value 30%? I think that this assumption has a high impact on PF ratio estimation.
- In statistical analysis section, there are a lot of statistical inaccuracies and I suggest to review this section with a statistician. For example, the statement “For descriptive statistics, data of both groups were analyzed using Fisher’s exact or Pearson’s X2 test for categorical data in cross-tabulation. Two group comparisons were performed by Mann-Whitney-U test.” is not exact because “..data of both groups were COMPARED using…” and “Mann-Whitney-U test was used to compare continuous variables between groups”. Moreover, the sentence “Data are given in tables as median and ranges”. What did the authors mean with the term “range” (interquartile range, minimum and maximum value)? Moreover, there are also percentages in the tables. Other example, the sentence “Cumulative incidences of postoperative pneumonia … were analyzed by Kaplan-Meier estimation and log-rank test.” Authors should declare that log-rank test was used to compare Kaplan-Meier curves.
- If PF ratio was not available (due to ICU discharge), authors assumed that this value is 300 mmHg and then they assessed correlation between PF ratio and type of surgery. I think that this assumption has a high impact on Spearman´s Rho rank correlation estimation.
- Authors enrolled 143 patients (105 OE and 38 LAE). They should explain how the study sample size was estimated.
Results.
- Values of relevant abdomino/thoracic extended procedures should be better reported in table 2.
- The first sentence in “Inflammation” section is not complete. Moreover, in the methods section, authors never mentioned that these data were collected. I suggest to insert this in Materials and methods section.
- In table 3, authors reported the number of missing values for leukocytes and C-reactive protein in each day of follow-up. I think that the number of missing values should be evaluated on the number of patients stayed in hospital at each day of follow-up. For this reason, it would be also useful to report the number of patients present in the hospital at each day of follow-up. Analogous comment for number of patients with PF ratio < 300 mmHg in table 5, where the comparison between groups should be done on percentages calculated on patients in hospital during each day of follow-up. Maybe, this was done from the authors but it is not clear.
- In table 4, authors reported postoperative hospital stay, stratifying in total, initial PO stay on ICU, cumulative PO stay on ICU. They should describe how this outcome was calculated in methods section.
- For me, it is not clear how authors evaluated the parameter “Pneumonia diagnosis on POD” reported in table 5.
- Authors detected a “not adjusted” difference between incidence of pneumonia and type of surgery. I think that they should consider to perform a multivariable analysis (e.g. Multivariable Cox regression models) in order to confirm their results, adjusting also for possible confounders. For example, there are some variables that differ between groups at baseline (prevalence of arterial hypertension, blood loss, duration of surgery, C-reactive protein at day 0) and they could be the reason of difference.
Discussion
- Limitations of the study, taking into account sources of potential bias or imprecision, were not reported.
Author Response
Reviewer #1
Title and Abstract.
- Study’s design is not reported in the title or the abstract. Authors just reported the terms “retrospective single center analysis of patients” but this does not identify a specific study type (cohort, case-control, cross-sectional,…).
- The study type (“retrospective cohort study”) is now defined in the abstract.
- I think that summary reported in the abstract is not very clear because authors reported many p-values without incidence values used for comparisons. Moreover, the meaning of acronym rsp was not reported.
- Thank you very much, the summary of the abstract was amended accordingly. The meaning of the acronym rsp was described on first mention.
- Authors should better report the parameter used to evaluate respiratory impairment. They used Horovitz index (PF ratio) less than 300 mmHg and not the oxygenation index (OI = FiO2/PaO2 * mean airway pressure).
- “Horovitz index” and “OI” were changed to “P/FR” = paO2/FiO2 ratio.
- In the keywords, I think the words “oxygenation index” and “acute respiratory syndrome” should be removed.
- These terms were deleted and the key words have been changed.
Introduction.
- Introduction is well written and I only suggest to remove the last sentence because it reports a result of the present study. I also suggest to call the Horovitz index as PF ratio or PaO2/ FiO2. This last comment is valid for the whole manuscript.
- Thank you for this comment. As in the abstract, “Horovitz index” or “OI” have been changed to “P/FR” = paO2/FiO2 ratio in the whole manuscript, as well as in the tables and the labeling of the Y-axis in Figure 2.
Materials and methods.
- The type of study design should be clearly reported.
- The study type (“retrospective cohort study”) had been defined in the material and methods section.
- Authors reported “For patients who were not mechanically ventilated (invasively or non-invasively), a FiO2 of 30% was anticipated.” Why did the authors decide to apply this assumption? Why the value 30%? I think that this assumption has a high impact on PF ratio estimation.
- Thank you for this important comment. We set the FiO2 to 0.3 for patients, who did not receive any ventilation (invasive or non-invasive), because most patients receive some kind nasal oxygen supply following extubation at the ICU. Unfortunately, the effect of nasal oxygen supply cannot be judged objectively. However, most of the calculated PF ratios were based on exact FiO2 values, obtained from the documentation of invasive or non-invasive ventilation. Nevertheless, this is certainly one of the limitations of the study, which are now included in the discussion.
- The same holds true for the estimation of PF ratio = 300 mm Hg for patients, who were classified as “normal oxygenation” for example on our normal ward (see below). A clinical criterion for a transfer to a regular hospital ward is sufficient respiration or oxygenation. Otherwise patients have to be referred to the ICU. From our clinical experience, patients with lower oxygen indices are symptomatic. As oxygenation impairment is defined in the literature by a PF ratio below 300 (according to the widely used Berlin definition of ARDS), we took this value as a threshold. Of course we might have underestimated these values, and this is now mentioned as a limitation of our study.
- In statistical analysis section, there are a lot of statistical inaccuracies and I suggest to review this section with a statistician.
- Our co-worker Florian Uhle has a qualification in statistics. He designed and reviewed the statistical analysis. However, we have done some revisions regarding the statistics – thank you for the comments!
For example, the statement “For descriptive statistics, data of both groups were analyzed using Fisher’s exact or Pearson’s X2 test for categorical data in cross-tabulation. Two group comparisons were performed by Mann-Whitney-U test.” is not exact because “..data of both groups were COMPARED using…” and “Mann-Whitney-U test was used to compare continuous variables between groups”.
- Thank you for this comment. We changed the statistical section regarding categorical and continuous
Moreover, the sentence “Data are given in tables as median and ranges”. What did the authors mean with the term “range” (interquartile range, minimum and maximum value)? Moreover, there are also percentages in the tables.
- We added the information, that we give continuous data in median and minimum to maximum range.
- n(%) indicate results of categorical data.
- For both, an additional comment has been included at the end of the materials and methods section.
Other example, the sentence “Cumulative incidences of postoperative pneumonia … were analyzed by Kaplan-Meier estimation and log-rank test.” Authors should declare that log-rank test was used to compare Kaplan-Meier curves.
- Thank you for this comment. This part of the statistics section has been changed.
- If PF ratio was not available (due to ICU discharge), authors assumed that this value is 300 mmHg and then they assessed correlation between PF ratio and type of surgery. I think that this assumption has a high impact on Spearman´s Rho rank correlation estimation.
- See above. This has been mentioned as a limitation of our study.
- Authors enrolled 143 patients (105 OE and 38 LAE). They should explain how the study sample size was estimated.
- As this is a retrospective analysis of patient data, the sample size is restricted to the patient collective, who was operated at our hospital during the time span of observation. Inclusion and exclusion criteria have been clearly stated within the material and methods section of the manuscript. Therefore, we included ALL patients, who underwent oncological abdomino-thoracic esophageal resections during the observational period and excluded patients, who for example underwent re-do esophageal resections, trans-abdominal distal esophageal resections et cetera.
Results.
- Values of relevant abdomino/thoracic extended procedures should be better reported in table 2.
- The values of extended procedures are now depicted more detail in Table 2.
- The first sentence in “Inflammation” section is not complete. Moreover, in the methods section, authors never mentioned that these data were collected. I suggest to insert this in Materials and methods section.
- The first sentence of the “inflammation” section was altered. A summary of the investigated main and secondary outcome parameters of the study is now included in the materials and methods.
In table 3, authors reported the number of missing values for leukocytes and C-reactive protein in each day of follow-up. I think that the number of missing values should be evaluated on the number of patients stayed in hospital at each day of follow-up. For this reason, it would be also useful to report the number of patients present in the hospital at each day of follow-up. Analogous comment for number of patients with PF ratio < 300 mmHg in table 5, where the comparison between groups should be done on percentages calculated on patients in hospital during each day of follow-up. Maybe, this was done from the authors but it is not clear.
- Please be aware of, that the minimum duration of postoperative hospital stay was 9 postoperative days. Patients, who were discharged or died, were not considered for the calculation in Table 3 because of lacking values. Therefore, those patients, who died, were indicated in the respective Table and the information regarding discharged patients is now included in the subheading of Table 3. Regarding Tables 4 and 5: only a small number of patients were discharged before POD 10 (n=2 in the OE and n=4 in the LAE group). Those patients who were discharged on POD 9 were considered as “no invasive ventilation”, “no catecholamine therapy”, “no respiratory impairment” for the calculations in Tables 4 and 5. Only patients, who died, were excluded from the calculation upon their death as indicated in Tables 4 and 5.
- In table 4, authors reported postoperative hospital stay, stratifying in total, initial PO stay on ICU, cumulative PO stay on ICU. They should describe how this outcome was calculated in methods section.
- As the duration of postoperative stay on ICU as well as total postoperative in-hospital stay are surrogate parameters of critical illness, frequency and severity of adverse events and complications, the duration of stay data were stratified into total postoperative in-hospital stay, initial postoperative stay at the ICU and cumulative postoperative stay on ICU. The latter was calculated for patients, who underwent repeated referrals to the ICU during the postoperative in-hospital stay. This is now mentioned in the material and methods section.
- For me, it is not clear how authors evaluated the parameter “Pneumonia diagnosis on POD” reported in table 5.
- As we described in the materials and methods, the pneumonia scoring system by Weijs et al. allows for a longitudinal evaluation of postoperative pneumonia based on infiltrates in chest radiography, body temperature and leukocyte counts. This is now clarified in legend to Table 5 and in materials and methods.
- Authors detected a “not adjusted” difference between incidence of pneumonia and type of surgery. I think that they should consider to perform a multivariable analysis (e.g. Multivariable Cox regression models) in order to confirm their results, adjusting also for possible confounders. For example, there are some variables that differ between groups at baseline (prevalence of arterial hypertension, blood loss, duration of surgery, C-reactive protein at day 0) and they could be the reason of difference.
- Thank you for this comment.
We agree that other factors might contribute to the incidence of pneumonia, we decided after thorough internal discussion against a multivariate analysis to avoid a statistical overstatement. Especially, the few differences observed - as correctly summarized by the reviewer - are either an inherent characteristic of the type of surgical approach (duration, C-reactive protein, blood loss) and therefore highly interdependent, or - as it is the case for arterial hypertension - without biological rationale, why this condition might increase the risk for pneumonia. Furthermore, the difference in case numbers might introduce a bias into the analysis.
However, the complete and elaborative evaluation of confounding factors for respiratory impairment and pulmonary complications after esophagectomy is the logical next step following our present results within the context of a larger sample size or prospective setting.
The main and most interesting results of this work (1. high rates of respiratory impairment in the early phase after esophagectomy, 2. differences between OE and LAE later on, lower pneumonia rates after LAE) are of high importance for our clinical routine patient care and for further studies regarding pulmonary complications and respiratory impairment after esophageal surgery. As we stated at the beginning of our discussion, this retrospective work is meant to create testable hypotheses.
In the present study we decided to use the Spearman´s rank correlation to evaluate influences of selected variables to perioperative PF ratios with a special focus on open or laparoscopic abdominal surgery during esophagectomy. The respective analysis was extended by the most important potential confounders (blood loss and intraoperative transfusion) upon the reviewer comments, but we are aware that this is not a comprehensive evaluation of potential confounding variables. Nevertheless, intraoperative blood loss is an important cause for intraoperative transfusion and therefore may influence perioperative pulmonary function.
Discussion.
- Limitations of the study, taking into account sources of potential bias or imprecision, were not reported.
- We have extended the discussion of the limitations of the study.

Reviewer 2 Report
Dear editors,
Thank you for considering me as a reviewer for this well written manuscript dealing with pulmonary complications and respiratory complaints after esophagectomy. Therefore, the authors describe a higher rate of pneumonia following open abdominothoracic esophagectomy compared with a hybrid minimally-invasive approach, which is well known from the current literature, but they go more into detail with evaluating perioperative oxygenation indexes, which had never been investigated before. Here, the authors found a surprisingly very high rate of respiratory impairment – equally in both groups – in the early postoperative phase after surgery. These are novel findings which bring new insights into the understanding and even pathophysiology of respiratory complaints and respiratory complication development after esophageal surgery.
Nevertheless, I have some comments, which should be considered prior to publication:
- Blood loss in the open surgery group was significantly higher (see Table 2). The authors should provide information regarding the frequency of the need of perioperative blood transfusion in both groups to estimate any kind of transfusion related lung injury (TRALI) as a source for early postoperative worse oxygenation indexes. This should be at least mentioned in the discussion of the manuscript.
- The findings by Babic and coworkers, very recently published (doi: 10.1016/j.athoracsur.2019.12.016.), may be useful for strengthen the discussion of high pneumonia rates within the scope of perioperative higher CRP values.
Author Response
Reviewer #2
(x) English language and style are fine/minor spell check required
- Some minor edits on English language have been done
Dear editors,
Thank you for considering me as a reviewer for this well written manuscript dealing with pulmonary complications and respiratory complaints after esophagectomy. Therefore, the authors describe a higher rate of pneumonia following open abdominothoracic esophagectomy compared with a hybrid minimally-invasive approach, which is well known from the current literature, but they go more into detail with evaluating perioperative oxygenation indexes, which had never been investigated before. Here, the authors found a surprisingly very high rate of respiratory impairment – equally in both groups – in the early postoperative phase after surgery. These are novel findings which bring new insights into the understanding and even pathophysiology of respiratory complaints and respiratory complication development after esophageal surgery.
- Thank you for these comments regarding our work.
Nevertheless, I have some comments, which should be considered prior to publication:
- Blood loss in the open surgery group was significantly higher (see Table 2). The authors should provide information regarding the frequency of the need of perioperative blood transfusion in both groups to estimate any kind of transfusion related lung injury (TRALI) as a source for early postoperative worse oxygenation indexes. This should be at least mentioned in the discussion of the manuscript.
- Although blood loss in the OE group was significantly higher, there were no differences in the percentage of patients, who received transfusion of blood products (fresh frozen plasma [FFP] and packed red blood cells) intraoperatively. Nevertheless, the proportion of patients, who received transfusion of blood products during their postoperative stay was higher in the OE group. To address the comment of the reviewer, we performed a novel rank correlation. Here, we could not see any significant inverse correlation between the rate of blood product transfusion (FFP + packed red blood cells) with the last intraoperatively assessed OI as well as oxygenation indexes on POD 0-3 in both groups, i.e. the postoperative phase where the cumulative incidences of worse OI rise rapidly. This was mentioned in the results section (see tables 2 and 6) and should be further investigated in more detail (amount of blood products, postoperative days of transfusion and respiratory impairment).
- The findings by Babic and coworkers, very recently published (doi: 10.1016/j.athoracsur.2019.12.016.), may be useful for strengthen the discussion of high pneumonia rates within the scope of perioperative higher CRP values.
- Thank you very much for this valuable comment. The main results by Babic et al. have been included in the discussion.

Round 2
Reviewer 1 Report
Thanks to the authors for their replies to my comments.
I am satisfied with the given answers except for one concerning the sample size of study population.
I asked to justify how the sample size had been estimated and for me it is not sufficient to say that the sample size is restricted to the patients operated at your hospital during the time span of observation. Authors should provide the minimum statistical detectable difference in outcome prevalence (pneumonia) and mean (PF ratio) that they can obtain performing an analysis on 143 patients (105 OE and 38 LAE).
Last comments. You decided after thorough internal discussion against a multivariate analysis to avoid a statistical overstatement. As authors explained me, there are good reasons for this and I think that this issue can be add in discussion.
Author Response
Reviewer #1
- Thanks to the authors for their replies to my comments.
- We have to thank the reviewers for their comments, which improved the manuscript.
- I am satisfied with the given answers except for one concerning the sample size of study population.
I asked to justify how the sample size had been estimated and for me it is not sufficient to say that the sample size is restricted to the patients operated at your hospital during the time span of observation. Authors should provide the minimum statistical detectable difference in outcome prevalence (pneumonia) and mean (PF ratio) that they can obtain performing an analysis on 143 patients (105 OE and 38 LAE).
- Based on this comment, we now included a power calculation for the comparison of postoperative pneumonia as well as cumulative reduced P/F ratio.
- Furthermore, based on our results given for the rate of postoperative pneumonia, we present the results of a sample size calculation, which could be used for future prospective studies, with a power of 0.8 and an alpha error probability of 0.05.
- Last comments. You decided after thorough internal discussion against a multivariate analysis to avoid a statistical overstatement. As authors explained me, there are good reasons for this and I think that this issue can be add in discussion.
- Thank you for this comment. We agree, thus we now included a statement, why not performing a multivariable analysis and why focusing on the given statistical analyses in the discussion section of the manuscript.